# Differential Expression of MED12-Associated Coding RNA Transcripts in Uterine Leiomyomas

**DOI:** 10.3390/ijms24043742

**Published:** 2023-02-13

**Authors:** Tsai-Der Chuang, Jianjun Gao, Derek Quintanilla, Hayden McSwiggin, Drake Boos, Wei Yan, Omid Khorram

**Affiliations:** 1Department of Obstetrics and Gynecology, Harbor-UCLA Medical Center, Torrance, CA 90502, USA; 2The Lundquist Institute for Biomedical Innovation, Torrance, CA 90502, USA; 3Department of Medicine, David Geffen School of Medicine at University of California, Los Angeles, Los Angeles, CA 90502, USA; 4Department of Obstetrics and Gynecology, David Geffen School of Medicine at University of California, Los Angeles, Los Angeles, CA 90502, USA

**Keywords:** leiomyoma, fibroid, myometrium, MED12 mutation, next-generation RNA sequencing

## Abstract

Recent studies have demonstrated that somatic MED12 mutations in exon 2 occur at a frequency of up to 80% and have a functional role in leiomyoma pathogenesis. The objective of this study was to elucidate the expression profile of coding RNA transcripts in leiomyomas, with and without these mutations, and their paired myometrium. Next-generation RNA sequencing (NGS) was used to systematically profile the differentially expressed RNA transcripts from paired leiomyomas (n = 19). The differential analysis indicated there are 394 genes differentially and aberrantly expressed only in the mutated tumors. These genes were predominantly involved in the regulation of extracellular constituents. Of the differentially expressed genes that overlapped in the two comparison groups, the magnitude of change in gene expression was greater for many genes in tumors bearing MED12 mutations. Although the myometrium did not express MED12 mutations, there were marked differences in the transcriptome landscape of the myometrium from mutated and non-mutated specimens, with genes regulating the response to oxygen-containing compounds being most altered. In conclusion, MED12 mutations have profound effects on the expression of genes pivotal to leiomyoma pathogenesis in the tumor and the myometrium which could alter tumor characteristics and growth potential.

## 1. Introduction

Leiomyomas, also known as uterine fibroids, are benign uterine tumors, characterized by excess accumulation of extracellular matrix (ECM), increased cell proliferation, and inflammation [1,2,3]. Ovarian steroids are known to promote the growth and progression of leiomyomas [4] which affect about 70% women in their reproductive years. There are higher prevalence and symptom severity in African Americans [5] compared with Caucasian women. The mechanisms underlying the pathogenesis of leiomyomas have been under intense investigation and recent evidence using microarrays and next-generation RNA sequencing has indicated aberrant expression of a host of protein-coding genes and non-coding RNA (ncRNA) in leiomyomas, including long and small non-coding RNA (lncRNA and sncRNA), which play crucial roles in regulation of protein-coding genes through post-transcriptional, transcriptional, and epigenetic mechanisms [6,7,8,9,10,11]. These tumors are also characterized by genetic heterogeneity with chromosomal re-arrangements, and mutations in a number of driver genes such as MED12 (mediator complex subunit 12), FH (fumarate hydratase), HMGA2 (high mobility group AT-hook 2), and COL4A5/6 (collagen type IV alpha 5 and alpha 6). These driver mutations have been associated with development and growth progression of leiomyomas [12,13,14,15]. Of the driver mutations, MED12 gene mutations have gained the most attention in the pathophysiology of leiomyomas [16,17,18,19]. MED12 is one component of the mediator complex, which in humans is composed of 30 subunits and functions as a transcription coactivator of RNA polymerase II [20]. It interacts with over 3000 transcription factors controlling the transcription of a wide array of genes [21]. MED12 along with MED13, cyclin C, and either CDK8 or CDK19 are components of the kinase module of this mediator. MED12 stimulates the kinase activity of CDK8 by bridging the interaction between MED13 and cyclin C–CDK8 [21]. Dysregulation of MED12 is found in many human cancers [21]. In leiomyoma, somatic MED12 mutations in exon 2 occur at a frequency of up to 80% regardless of race/ethnicity [16,17,18,19]. The MED12-mutant leiomyomas are smaller in size and more often subserous in location [22]. Mutations in exon 2 of MED12 are gain-of-function mutations, as evidenced by transgenic mice which developed leiomyoma-like tumors in the uterus when a conditional mutation in MED12 was introduced [23]. Mutations in the MED12 gene have been shown to disrupt its ability to activate cyclin C-dependent CDK8 [24], and its CDK19 stimulatory activity [25]. Moreover, these mutations were reported to be associated with abnormal activation of many genes regulating important pathways in leiomyoma pathogenesis, including the Wnt/β-catenin signaling, hedgehog signaling, sex steroid receptor signaling, and transforming growth factor (TGF)-β receptor signaling pathways [17,18,19,26,27]. 

The impact of the high occurrence of MED12 mutations in leiomyomas on the expression of genes critical to leiomyoma pathogenesis remains unknown. Therefore, the objective of the present study was to determine and compare the expression profile of differentially expressed coding RNA transcripts using high-throughput sequencing and in-depth data analysis based on the MED12 mutation status of the leiomyoma. This next-generation sequencing (NGS) profiling was followed by validation studies of 31 differentially expressed genes in 73 specimens using quantitative reverse-transcription polymerase chain reaction (qRT-PCR).

## 2. Results

### 2.1. High-Throughput Sequencing of Coding RNA Transcripts in Leiomyoma and Matched Myometrium

We initiated our investigation with high-throughput next-generation RNA sequencing (NGS) with RNA isolated from nineteen paired leiomyomas, including eight MED12-mutation-negative and eleven MED12-mutation-positive leiomyomas. We first analyzed the differential expression of genes in all the specimens regardless of MED12 mutation analysis. Following normalization of 29,354 RNA transcripts, hierarchical clustering and TreeView analysis, comparing leiomyomas with the matched myometrium group, resulted in identification of 9699 RNA transcripts whose expression was altered significantly, with overexpression of 5665 RNA transcripts and downregulation of 4034 RNA transcripts by 1.5-fold or greater in leiomyoma (Figure 1A). Volcano plot filtering indicated that 2344 RNA transcripts were upregulated, and 1711 RNA transcripts were downregulated in leiomyomas compared with their matched myometrium (fold change > 1.5 and *p* < 0.05; Figure 1B). Principal component analysis (PCA), which examines the similarity in the pattern of RNA transcripts among two groups, was performed, followed by k-mean analysis to cluster the samples, indicating the high reliability of the data (Figure 1C). Subsequently, by using the STRING database and Cytoscape software a functional analysis of the protein–protein interaction (PPI) network was constructed demonstrating the highlighted associations between target genes involved in leiomyoma pathogenesis (Figure 1D). The involvement of many proteins such as FN1, COL1A1, BMP7, CCND1, EZH2, HMGA2, AhR, and E2F1 shown in the protein–protein interaction networks are well known in leiomyoma pathogenesis [8,28,29,30,31,32]; others, such as SOX2, REN, PPARG, ABCB1, and NR3C1 are novel and require further investigation.

### 2.2. Differential Expression of MED12-Associated Coding RNA Transcripts in Leiomyoma and Matched Myometrium

We then examined the differential expression of genes in MED12-mutated and non-mutated specimens. We observed a significant overlap of differentially expressed genes between the two groups although in mutated specimens the degree of change was greater for most genes. The analysis was performed as fold change (leiomyoma/paired myometrium) comparing mutated and non-mutated group. This analysis, based on differential expression, resulted in identification of 2757 RNA transcripts with altered expression, of which the expression of 1406 RNA transcripts was increased, while the expression of 1351 RNA transcripts was decreased by 1.5-fold or greater in MED12-mutated specimens compared with non-mutated specimens. Hierarchical clustering and TreeView analysis separated these transcripts into their respective groups (Figure 2A). We identified 394 transcripts that showed more than 1.5-fold change (up or down) in the mutated specimens but not in the non-mutated specimens. The heat map (Figure 2B) shows 109 out of the 394 transcripts that were uniquely altered in the MED12-mutated specimens. The gene ontology (GO) and KEGG (Kyoto Encyclopedia of Genes and Genomes) pathway enrichment analyses for these 109 transcripts revealed that the genes that were uniquely expressed in the mutated group were predominantly involved in the regulation of extracellular matrix constituents (Figure 2C).

### 2.3. Differential Expression of MED12-Associated Coding RNA Transcripts in Myometrium

We next sought to examine if there were any differences in the expression of genes in the myometrium of specimens with the MED12 mutation compared with non-mutated specimens. Although we did not detect mutations in the MED12 gene in any myometrial tissues, we observed marked differences in the expression profile in the myometrium adjacent to MED12-mutated leiomyomas compared with the myometrium adjacent to MED12 wild-type tumors. The analysis of RNA sequencing data in myometrium based on MED12 mutation status of their paired adjacent leiomyomas indicated that the expression of 6259 RNA transcripts were altered, of which 2219 RNA transcripts were up-regulated, while 4040 RNA transcripts were downregulated by 1.5-fold or greater in myometrium adjacent to MED12-mutated leiomyomas compared to myometrium adjacent to MED12 wild-type leiomyomas. Hierarchical clustering and TreeView analysis separated these transcripts into their respective group (Figure 3A). A volcano plot, a PCA and a PPI network were constructed to highlight MED12-associated RNA transcripts in their paired adjacent leiomyomas (Figure 3B–D). Many proteins such as F3, WNT2, CDH1, and LIF [33,34,35,36] shown in the protein–protein interaction networks are well known in leiomyoma pathogenesis, and others, such as ICAM1, CXCL8, CCL2, and NANOG are novel and require further investigation. The gene ontology (GO) and KEGG pathway enrichment analyses of genes with altered expression in the myometrium indicated that these genes were mainly involved in the response to oxygen-containing compounds and the extracellular space (Figure 3E). Selected genes in each of the pathways shown in Figure 3E are presented in Appendix A. We selected five of the differentially expressed novel RNA transcripts in the myometrium (*WNT16*, *CACNA1D*, *DCX*, *MTMR8*, and *WIF1*) for confirmation using qRT-PCR in seventy-three paired specimens (Figure 4). This analysis indicated that the expression of *WNT16*, *CACNA1D*, *DCX*, and *WIF1* was significantly higher, while the expression of *MTMR8* was significantly lower in myometrium adjacent to MED12-mutated leiomyomas compared to myometrium adjacent to MED12 wild-type leiomyomas (Figure 4).

### 2.4. Validation of MED12-Associated Coding RNA Transcripts in Leiomyoma and Matched Myometrium

To provide support and validate the NGS data we selected 31 differentially expressed coding RNA transcripts for confirmation studies using qRT-PCR (n = 73) including the same tissues used for RNA sequencing (Figure 5). Of the 73 paired specimens, 48 leiomyomas had MED12 mutations (65.8%). The prevalence of MED12 mutations in our patient population was in agreement with previously published reports [16,17,18,19]. The gene ontology (GO) and KEGG pathway enrichment analyses of the selected differentially expressed RNA transcripts revealed that these genes were involved in multiple signaling pathways such as ECM–receptor interaction (Figure 5A; *COL11A1*, *MUC12*, *FRMD5*, *FCGBP*, *EGFL6*, *FN1*, *KLK5*, *ITGA9*, and *DCX*), cell signaling pathways including the Wnt signaling pathway (Figure 5B; *WNT4*, *WNT2*, *WNT16*, and *WIF1*), chemokine signaling pathways (Figure 5B; *CXCL13*), PI3K/AKT signaling (Figure 5B; *CBX8*), TLR signaling (Figure 5B; *S100A1*), signaling by Rho GTPases (Figure 5B,C; *NDC80* and *NTM*), the VEGFA–VEGFR2 signaling pathway (Figure 5C; *RAB37*), signal-transducing phosphatase (Figure 5C; *PPP1R14C*), the Jak/STAT signaling pathway (Figure 5C; *JAK3*), the TGF-β signaling pathway (Figure 5C; *BMP7*), transcription regulation (Figure 5C; *S100A4* and *HMX1*), ion homeostasis (Figure 5C; *CACNA1D* and *RGS4*), and metabolic pathways (Figure 5D; *EZH2*, *IGFBPL1*, *MTMR8*, *ATP5MC1P1*, and *CYP19A1*). Among the 31 RNA transcripts validated, the expression of *COL11A1*, *MUC12*, *FRMD5*, *FCGBP*, *EGFL6*, *FN1*, *KLK5*, *ITGA9*, *DCX*, *WNT4*, *WNT2*, *WNT16*, *WIF1*, *CXCL13*, *CBX8*, *NDC80*, *NTM*, *PPP1R14C*, *BMP7*, *EZH2*, *IGFBPL1*, *MTMR8*, *ATP5MC1P1*, *CYP19A1*, *HMX1*, *CACNA1D*, and *RGS4* was significantly higher, while the expression of *S100A1*, *RAB37*, and *S100A4* was significantly lower in leiomyomas compared to matched myometrium (Figure 5A–D). Further comparisons were made comparing mutated leiomyomas to non-mutated leiomyomas and mutated myometrium to non-mutated ones (Figure 6A–D). The differentially expressed genes (leiomyoma/paired myometrium) in the two comparison groups are shown in Figure 7. As shown in Figure 6, the magnitude of differences in expression between the mutated and non-mutated groups was most pronounced for *KLK5*, *WNT 4*, *WNT16*, *PPP1R14C*, *NTM*, *IGFBPL1*, *EGFL6*, *FCGBP*, *COL11A1*, *FRMD5*, *DCX*, *CBX8*, and *RAB37*. A summary of the analyses from Figure 6 and Figure 7 is shown in Table 1. The expression pattern of these 31 genes when divided into mutated and non-mutated groups, was similar to the analysis when all samples were combined regardless of mutation status (Figure 5), with the exception of *JAK3*, which showed no significant difference in expression in the combined group analysis but was significantly different when analyzed based on the mutation status (Figure 5C). The expression of the 31 RNA transcripts was also similar when the data were analyzed as differentially expressed versus when the comparison was performed based on expression levels in the leiomyomas alone, except for *S100A1*, *MTMR8*, *CACNA1D*, *DCX*, *BMP7*, *RGS4*, and *CYP19A1* (Table 1).

## 3. Discussion

The results of this study provide a comprehensive profile of protein-coding genes whose expression is altered by the presence or absence of MED12 mutations in the tumor. Although there was a significant overlap of differentially expressed genes in mutated and non-mutated specimens, there were 394 transcripts that were uniquely aberrantly expressed in the mutated specimens compared with non-mutated ones. Using the DAVID gene functional classification tool [73,74] for gene ontology (GO) and KEGG pathway enrichment analyses for these transcripts indicated that these genes were predominantly involved in the regulation of extracellular matrix constituents, response to external stimulus, and extracellular space. Although the myometrium of mutated specimens did not express MED12 mutations, the expression of *WNT16*, *CACNA1D*, *DCX*, *WIF1*, and *MTMR8* genes were significantly altered in the myometrium of MED12-mutated specimens compared with non-mutated specimens, with *WNT16*, *CACNA1D*, *DCX*, and *WIF1* genes upregulated and the *MTMR8* gene downregulated. Pathway enrichment analysis for these myometrial genes showed predominant involvement in response to oxygen-containing compounds, extracellular space, and response to organic substances and external stimuli. Validation studies using PCR, performed in 73 specimens, confirmed the RNA-seq results indicating the differential expression of *COL11A1*, *MUC12*, *FRMD5*, *FCGBP*, *EGFL6*, *FN1*, *KLK5*, *ITGA9*, *WNT4*, *WNT2*, *WNT16*, *CXCL13*, *CBX8*, *NDC80*, *NTM*, *PPP1R14C*, *JAK3*, *EZH2*, *IGFBPL1*, *MTMR8*, *ATP5MC1P1*, *HMX1*, *CYP19A1*, *RGS4*, *BMP7*, *DCX*, and *CACNA1D* mRNA was significantly higher, while the expression of *S100A1*, *RAB37*, and *S100A4* mRNA was significantly lower in MED12-mutated leiomyomas compared with non-mutated specimens. 

The mediator complex is composed of the head, middle, tail, and kinase modules and gene-specific transcription factors bind through the tail and kinase domains and then transduce the regulatory information through the middle and head modules to RNA polymerase 11, thereby regulating the expression of a wide array of genes [21]. It remains to be determined how, despite the loss of mediator-associated CDK activity [24], the transcription of so many genes as shown here are altered by the MED12 mutations. One possibility for the widespread effects of these mutations on gene expression is that the mediator is known to interact with super enhancers which regulate gene families [75], and in a recent publication [11] we reported that the expression of multiple super enhancers is altered in leiomyomas, many of which were dependent on the MED12-mutation status of the tumors. Our data indicate that the presence of a MED12 mutation does not only affect the expression of genes in the leiomyoma but also in the myometrium which does not express the MED12 mutation. Potential mechanisms that could account for this crosstalk between the leiomyomas and the adjacent myometrium is through exosomes containing regulatory RNAs, the expression of which are altered in the mutated leiomyomas [6,76]. These regulatory RNAs such as miRNAs within the exosome could, through the systemic circulation or in a paracrine manner, influence the expression of genes in the myometrium. The myometrial genes whose expressions were altered by the presence of MED12 mutations were predominantly involved in processes related to response to oxygen-containing compounds, extracellular space, and response to external stimuli. The importance of the myometrium in the pathophysiology of leiomyomas was highlighted in recent reports showing myometrial oxidative stress can drive MED12 mutations in leiomyomas [77], and another report demonstrating differences in the transcriptome of myometrium from patients with leiomyomas compared with myometrium from non-diseased uteri [78]. In addition, uterine contractions induce uterine hypoxia in reproductive-aged women [79], and this insult can cause activation of NF-κB-mediated inflammation as has been demonstrated for human stromal cells [80].

WNT/β-catenin plays a pivotal role in leiomyoma pathogenesis. In an early study, constitutive activation of β-catenin led to development of mesenchymal tumors similar to leiomyomas in mice [81]. Another study demonstrated that WNT4 was significantly overexpressed in leiomyoma intermediate cells, inducing cell proliferation through Akt-dependent β-catenin activation, and resulting in the induction of pro-proliferative genes such as cyclin D1 and c-Myc [82]. Moreover, the expression of WNT16 in leiomyoma stem cells was increased in response to estrogen and progesterone and led to growth of leiomyoma cells in a paracrine manner through activation of canonical WNT signaling [83]. WNT2, WNT4, and WNT16 are subunits of the WNT gene family and have been reported to be involved in several developmental processes and in tumorigenesis [50]. Here we showed that WNT2, WNT4, and WNT16 are overexpressed in leiomyomas, and that this upregulation was more pronounced in MED12-mutated leiomyomas compared to MED12 mutation-negative specimens, which is in line with previous reports [17,19,84]. In addition, the myometrial expression of WNT16 in MED12-mutated specimens was greater compared to myometrium of mutation-negative leiomyomas. These findings raise the intriguing possibility that MED12 mutations in the leiomyomas also induce changes in the expression of WNT16 in the adjacent myometrium, which further fuels tumorigenesis in mutated specimens; this possibility could account for the observation of multiplicity of tumors in MED12-mutated specimens compared to non-mutated tumors which are generally solitary [22]. An unexpected finding was the overexpression of the WNT inhibitory factor (WIF1) in leiomyomas. WIF1, is a lipid-binding protein that binds to WNT proteins and prevents them from signaling [85]. The presence of mutations did not affect the degree of overexpression of WIF1; however, the myometrium of mutated specimens also overexpressed *WIF1* mRNA compared to myometrium of non-mutated specimens. Others have also reported overexpression of WIF1 and other WNT-protein inhibitors such as SFRP1, FBXW11, NKD1, and SFPR4 in leiomyomas [84]. Moreover, the expression of DKK1 and SFRP1 was higher in MED12-mutated tumors compared with non-mutated ones [84]. A potential possibility is that overexpression of these WNT-inhibitory proteins could be a mechanism to prevent leiomyomas from becoming malignant and invasive. 

One of the hallmarks of leiomyomas is the accumulation of ECM [2]. Our profiling indicates that the genes that were uniquely dysregulated in MED12-mutated specimens predominantly fell in the category of ECM structural constituents, among them were *KRT80*, *COL5A3*, *NDC80*, *WNT4*, *MTMR8*, and *FGF17* (Figure 2B). Several other genes, although upregulated in non-mutated samples, showed a greater magnitude of increase in expression in the mutated leiomyoma group, including *FN1*, *DCX*, and *COL11A1*. FN1 is a major component of the extracellular matrix and exists as a dimeric or multimeric form linked by disulfide bonds to other extracellular matrix proteins such as integrins, collagen, fibrin, and heparan sulfate proteoglycans [44,45]. FN1 plays a major role in cell adhesion, growth, differentiation, and migration processes including wound healing and embryonic development, host defense, blood coagulation, and metastasis [44,45]. Several groups have reported that leiomyomas express higher levels of FN1 compared to myometrium [8,86]. The greater expression of FN1 in mutated leiomyomas suggests a much greater accumulation of ECM in mutated tumors, as FN1 is one of the major components of ECM. A recent report showed increased ECM activity in MED12-mutated tumors and although no specific ECM genes were specified, histologic evidence for increased collagen levels in mutated tumors was provided. The authors of this study also provided evidence for differences in the DNA methylome of mutated tumors, although no validation studies were provided for any specific genes [87]. Another component of the ECM is COL11A1 which is one alpha chain of type XI collagen and is involved in fibrillogenesis through regulation of the lateral growth of collagen II fibrils [37]. Recent evidence has shown that COL11A1 is frequently overexpressed in various tumors and its expression is highly correlated with tumor aggressiveness. Our results indicate greater overexpression of COL11A1 in mutated tumors compared with non-muted ones. DCX is another component of the ECM. It is a microtubule-associated protein belonging to the doublecortin family. DCX is expressed in immature neurons and neuronal precursor cells and binds to the microtubule cytoskeleton to direct neuronal migration through modulation and stabilization of microtubules [49]. Several studies suggested that DCX is correlated to cancer cells invasion and metastasis and is overexpressed in leiomyomas as well [49,88,89]. Our data indicate that in addition to marked overexpression of DCX in leiomyomas, in both mutated and non-mutated tumors with greater increase in the mutated leiomyomas, the myometrium of MED12-mutated specimens also overexpressed DCX. This finding points to a probable significance of DCX in ECM remodeling and crosstalk between leiomyoma and myometrium in this process. Another noteworthy ECM-regulating gene is KLK5 which was differentially overexpressed in mutated leiomyomas but not in the non-mutated specimens. KLK5 is a serine protease and a member of the kallikrein subfamily with involvement in collagen formation and MSP-RON signaling [46,47]. Growing evidence suggests that KLK5 is upregulated in numerous types of cancers such as colorectal cancer, gastric cancer, and oral cancer and is implicated in carcinogenesis [90,91,92,93]. KLK5 is upregulated by estrogens and progestins [46]. One substrate of KLK5 is PAR2, which triggers pro-inflammatory mediators and induces the expression of NF-κB target genes in both immune cells and tumor cells [91]. Our group and others have previously reported on the significance of inflammation and the NF-κB pathway in leiomyoma pathogenesis [94,95]. Overall, the differential expression of genes regulating the constituents of ECM in mutated and non-mutated specimens suggests potential differences in tumor stiffness in mutated leiomyomas compared with non-mutated leiomyomas and merits further investigation.

Our data indicate that the expression of a number of genes regulating calcium homeostasis is dysregulated in leiomyomas in a MED12-dependent manner. Among these genes are the calcium-binding proteins S100A4 and S100A1, and CACNA1D (calcium voltage-channel subunit alpha-1D) which regulates calcium entry into cells [53,54,69,71]. S100A1, upon binding to calcium, undergoes conformational change and interacts with numerous protein targets such as those involved in calcium signaling, filament-associated proteins, transcription factors, and others [96,97]. S100A4 plays a vital role in fibrotic diseases [98]. It is released at inflammatory sites and is implicated in ECM remodeling, cell motility, invasion, and angiogenesis [98] and in epithelial–mesenchymal transition [99] all of which are critical in leiomyoma pathogenesis. In the realm of cancer, S100A4 is a transcriptional target of β-catenin and promotes tumor migration, invasion, and metastasis [100]. Our data, demonstrating decreased expression of S100A4 and S100A1 particularly in mutated tumors and the increase in CACNA1D expression, suggest potential reduced binding of calcium and greater calcium entry into leiomyoma cells which could contribute to tumor calcification. Currently there are no data indicative of a difference in calcium content of mutated compared with non-mutated tumors. 

The growth of leiomyomas is dependent on estrogen [4] and early studies demonstrated that the expression level of CYP19A1, or aromatase, which is a member of the cytochrome P450 superfamily and converts androgens to estrogens [68], is higher in leiomyomas [101] and this increase is more pronounced in African American women compared with Japanese women [102]. Here we confirmed that leiomyomas overexpress aromatase, but, more importantly, we showed that this increase is of greater magnitude in MED12-mutated tumors compared with non-mutated tumors. Leiomyomas are also characterized by a number of epigenetic modifications [103]. Among the epigenetic enzymes involved in these modifications is EZH2 which is a catalytic core protein in PRC2 and catalyzes the methylation of H3K27, thereby silencing the transcription of target genes [64]. Several studies demonstrated overexpression of EZH2 in leiomyomas [104,105]. Here we confirmed higher expression of EZH2 in leiomyomas and showed that the magnitude of overexpression is higher in MED12-mutated tumors compared with non-mutated tumors. Collectively, these results indicate potential greater local exposure to estrogen in mutated tumors along with greater degree of epigenetic modifications. 

In summary, we demonstrate that although there is a significant overlap in the differential expression of genes in MED12-mutated and non-mutated leiomyomas, the degree of change in expression is heightened for many genes in the MED12-mutated specimens. One limitation of our study is that a myomatous uterus has multiple tumors which might express different types of MED12 mutations [32], and these mutations may not induce the same changes in the transcriptome. Furthermore, a whole host of genes are uniquely over- or under-expressed in mutated tumors. These unique genes primarily regulate ECM structural constituents and components, and response to external stimuli. Our study also points to the significance of the myometrial compartment as a site of gene dysregulation as evidenced by differences in the expression of a unique set of genes in the myometrium of MED12-mutated specimens compared with non-muted myometrium. This result suggests a potential communication between the tumor and the adjacent myometrium which could contribute to tumor progression and growth. 

## 4. Materials and Methods

### 4.1. Myometrium and Leiomyoma Tissue Collection 

Portions of uterine leiomyomas (intramural, 3–5 cm in diameter) and paired myometrium (n = 73) were obtained from patients at Harbor-UCLA Medical Center. Prior approval from the Institutional Review Board (18CR-31752-01R) at the Lundquist Institute was obtained. Informed consent was obtained from all patients participating in the study who were not taking any hormonal medications for at least 3 months prior to surgery. The paired tissues were from Caucasian (n = 9), African American (n = 23), Hispanic (n = 37), and Asian (n = 4) women aged 30–54 years (mean 45 ± 5.2 years). The tissues were snap-frozen and stored in liquid nitrogen for further analysis as previously described [33,35]. 

### 4.2. MED12-Mutation Analysis 

Genomic DNA from leiomyomas and paired myometrial specimens was extracted from 100 mg of freshly frozen tissue using MagaZorb DNA Mini-Prep Kits (Promega, Madison, WI, USA) according to the manufacturer’s protocol. PCR amplification and Sanger sequencing (Laragen Inc., Culver City, CA, USA) were performed to investigate MED12 exon 2 mutations using the following primer sequences in the 5′–3′ direction: sense, GCCCTTTCACCTTGTTCCTT and antisense, TGTCCCTATAAGTCTTCCCAACC. PCR products were sequenced using Big Dye Terminator v.3.1 sequencing chemistry and the sequences were analyzed with the Software ChromasPro 2.1.8 and compared with the MED12 reference sequence (NG_012808 and NM_005120). The 19 pairs of tissues used for next-generation RNA sequencing were from 11 MED12-mutation positive and 8 MED12-mutation negative leiomyomas. The mutation analysis of the specimens (n = 73) indicated that 48 leiomyomas had MED12 mutations (48/73 pairs; 65.8%) with no mutations in the myometrium. Missense mutations in MED12 exon 2 were the most frequent alteration (39/48 pairs), followed by in-frame insertion–deletion type mutations (9/48 pairs). The missense mutations in exon 2 included c.130G > C (p.Gly44Arg) (5/39 pairs), c.130G > A (p.Gly44Ser) (7/39 pairs), c.130G > T (p.Gly44Cys) (2/39 pairs), c.131G > C (p.Gly44Ala) (2/39 pairs), c.131G > A (p.Gly44Asp) (17/39 pairs), c.131G > T (p.Gly44Val) (5/39 pairs), and c.128A > C (p.Gln43Pro) (1/39 pairs).

### 4.3. RNA Sequencing and Bioinformatic Analysis

Total RNA was extracted from leiomyomas and matched myometrium using TRIzol (Thermo Fisher Scientific Inc., Waltham, MA, USA). RNA concentration and integrity was determined using a Nanodrop 2000c spectrophotometer (Thermo Scientific, Wilmington, DE, USA) and Agilent 2100 Bioanalyzer (Agilent Technologies, Santa Clara, CA, USA) as previously described [9,29,30,106]. Samples with RNA integrity numbers (RIN) greater than or equal to 9 were used for library preparation. One microgram of total RNA from each tissue was used to produce strand-specific cDNA libraries using Truseq (Illumina, San Diego, CA, USA) according to manufacturer’s instructions. The RNA sequencing was carried out at Novogene Corporation Inc. The computational analysis was started by establishment of pairs of Fastq files for all samples. The workflow consisted of QC check (FastQC) -> alignment (Hisat2) -> feature counts (subread) -> differential gene expression analysis (DESeq2). For quality control FastQC was used to check the quality of raw fastq data from sequencing core and after adaptor cut and quality trimming [107]. FastQC reports consisted of multiple parameters such as the number of reads, duplicates, adapter contents, and sequence quality score. HISAT2 was applied to perform alignment [108]. The reads of raw fastq data with either end were distributed from 17.1 M to 34.8 M and the alignment rates were higher than 95%. Assigned reads for feature count were distributed from 17.7 M to 36.7 M in correspondence to the rate of assigned reads from 80.1% to 96.5%. Each sample produced a bam file after alignment to the genome. Features from each bam file that mapped to the genome in the provided annotation file were counted using the subread function [109]. MultiQC was used to analyze and integrate the QC reports, with input data from reports of fastqc, alignment reads, and feature assigned [110]. An R package, DESeq2, was used to analyze differential gene expression [111]. Markers/genes with sum-of-read counts across all cases and controls that were 10 or greater were kept for the downstream analysis. During DGE analysis, a boxplot of the Cook’s distances was prepared to determine if any sample raw reads departed from others. All our samples’ raw reads in the boxplot were relatively even and all samples were included into the downstream analysis for differential gene expression (DGE). To visualize the strength of differential gene expression, the hierarchical clustering and TreeView graphs, volcano plots, PCA plots, and pathway enrichment analysis plots were prepared using Flaski [112], and the protein–protein interaction networks were prepared using the Search Tool for the Retrieval of Interacting Genes (STRING) database [113]. Overall, all differential gene expressions were acceptable for subsequent statistical analysis. The RNA sequencing data were deposited in Gene Expression Omnibus (GEO) database with accession number (GSE224991).

### 4.4. Quantitative RT-PCR 

Briefly, 2 μg of RNA was reverse transcribed using random primers for selected genes according to the manufacturer’s guidelines (Applied Biosystems, Carlsbad, CA, USA). Quantitative RT-PCR was carried out using SYBR gene expression master mix (Applied Biosystems). Reactions were incubated for 10 min at 95 °C followed by 40 cycles for 15 s at 95 °C and 1 min at 60 °C. The expression levels of selected genes were quantified using the Invitrogen StepOne System with FBXW2 (F-box and WD repeat domain containing 2) used for normalization [114]. All reactions were run in triplicate and relative mRNA expression was determined using the comparative cycle threshold method (2-ΔΔCq), as recommended by the supplier (Applied Biosystems). Values were expressed as fold change compared to the control group. The primer sequences in the 5′–3′ direction used are listed in Appendix A.

### 4.5. Statistical Analysis

Throughout the text, results are presented as mean ± SEM and were analyzed using PRISM software (Graph-Pad, San Diego, CA, USA). Dataset normality was determined using the Kolmogorov–Smirnoff test. The data presented in this study were not normally distributed and therefore non-parametric tests were used for data analysis. Comparisons involving two groups were analyzed using the Wilcoxon matched pairs signed rank test (Figure 5) or the Mann–Whitney test (Figure 4, Figure 6 and Figure 7) as appropriate. Statistical significance was established at *p* < 0.05.

## Figures and Tables

**Figure 1 ijms-24-03742-f001:**
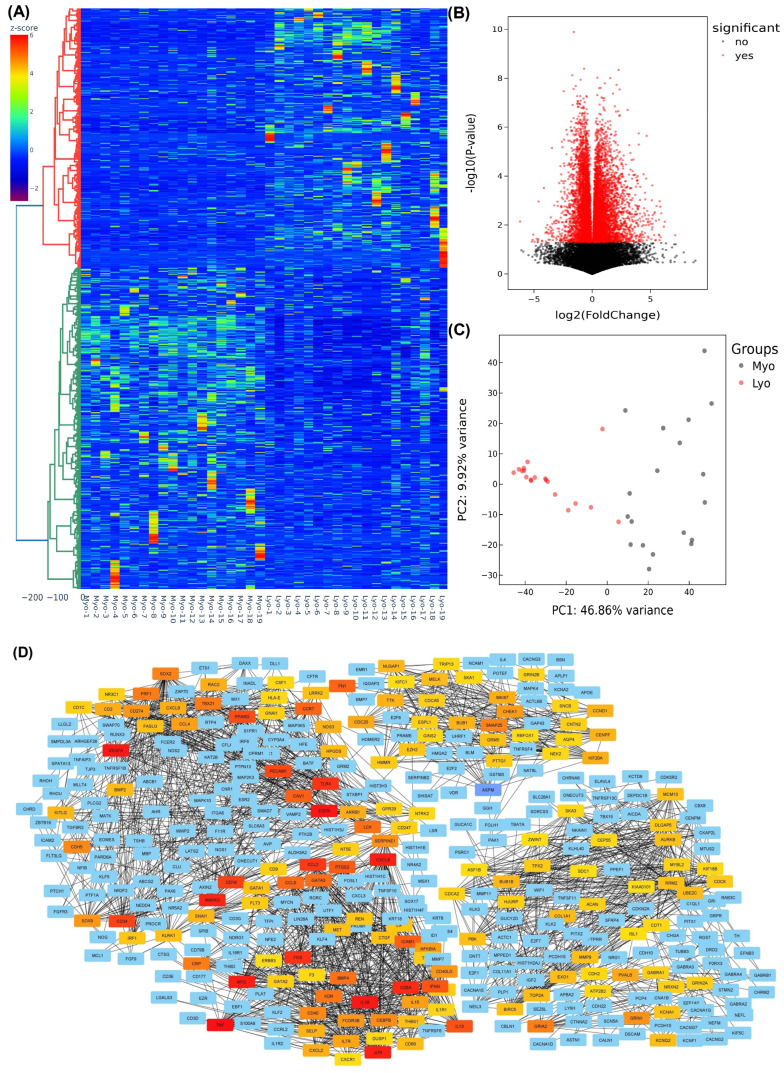
Heterogeneity and transcriptomic changes in leiomyomas compared to the myometrium. (**A**) Hierarchically clustered heatmap analysis of the differentially expressed transcripts (fold change ≥ 1.5, *p* < 0.05) in 19 paired leiomyomas and matched myometrium. Color gradient represents gene expression as z-scores. (**B**) Volcano plot showing up- (n = 2344) and downregulated genes (n = 1711) with a false discovery rate (FDR), *p*-value < 0.05, depicted as red dots. (**C**) Principal component analysis (PCA) plot of RNA-seq results from paired leiomyomas and matched myometrium (n = 19). Each dot represents one sample. Myometrial samples (Myo) are shown in black and leiomyoma samples (Lyo) are shown in red. (**D**) The protein–protein interaction networks were constructed with the Search Tool for the Retrieval of Interacting Genes (STRING) database and Cytoscape software version 3.9.1 using the 150 hub genes identified by the CytoHubba plugin of the Cytoscape software platform. The color of nodes denotes interaction degree (from high to low degree: red, orange, yellow, and blue).

**Figure 2 ijms-24-03742-f002:**
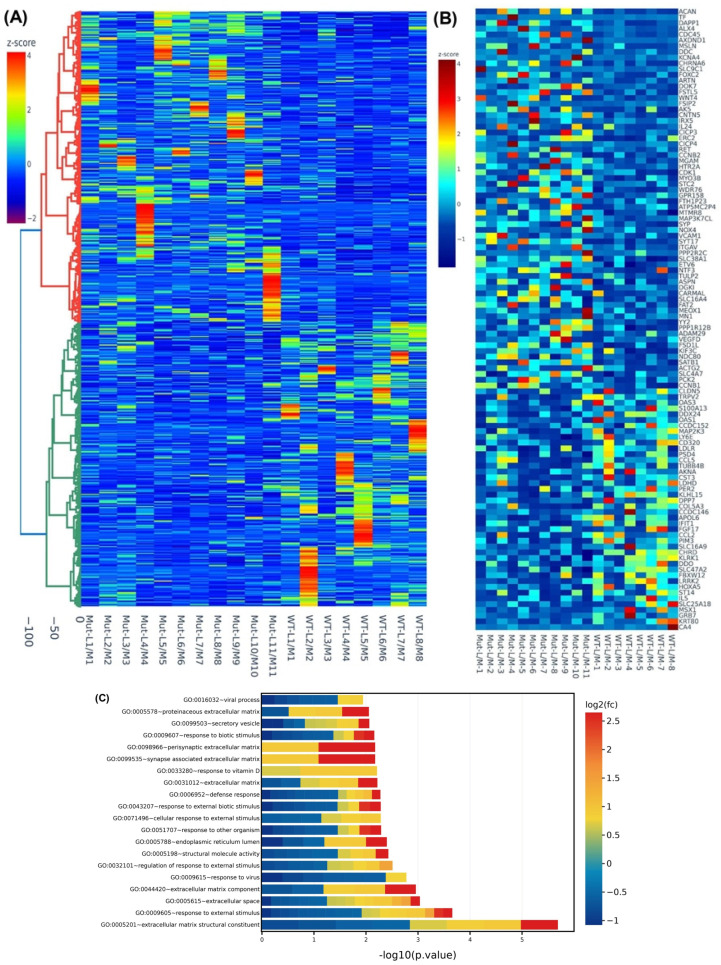
Differential levels of transcriptomic heterogeneity of genes in paired specimens analyzed based on MED12-mutation status. (**A**) Hierarchically clustered heatmap analysis was performed as fold change (leiomyoma/paired myometrium) comparing the MED12-mutated (n = 11) with the non-mutated (n = 8) group (fold change ≥ 1.5, *p* < 0.05). Color gradient represents gene expression as z-scores. (**B**) Heatmap of the 109 enriched genes (leiomyoma/paired myometrium) in the MED12-mutated (n = 11) but not in the non-mutated (n = 8) group (fold change ≥ 1.5, *p* < 0.05). Color gradient represents gene expression levels as z-scores. (**C**) Gene ontology (GO) analysis of 394 differentially expressed genes (leiomyoma/paired myometrium) in the MED12-mutated (n = 11) but not in the non-mutated (n = 8) group (fold change ≥ 1.5, *p* < 0.05). Color gradient represents levels of log2 fold change presented as z-scores.

**Figure 3 ijms-24-03742-f003:**
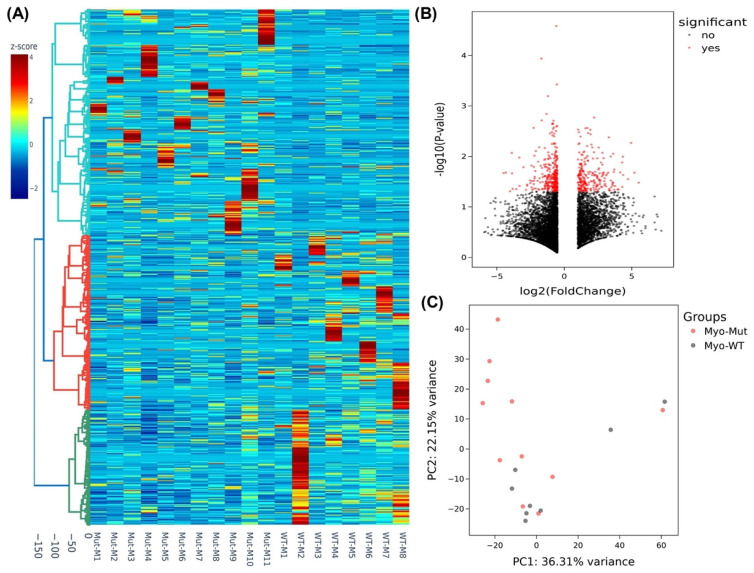
Heterogeneity and transcriptomic changes in myometrium specimens based on MED12-mutation status of their paired adjacent leiomyomas. (**A**) Hierarchically clustered heatmap analysis was performed in the myometrium comparing the MED12-mutated (n = 11) with the non-mutated (n = 8) group of their paired adjacent leiomyomas (fold change ≥ 1.5, *p* < 0.05). Color gradient represents gene expression as z-scores. (**B**) Volcano plot showing up- (n = 2219) and downregulated genes (n = 4040) with a false discovery rate (FDR), *p*-value < 0.05, depicted as red dots. (**C**) Principal component analysis (PCA) plot of RNA-seq results in the myometrium comparing the MED12-mutated (n = 11) with non-mutated (n = 8) group of their paired adjacent leiomyomas. Each dot represents one sample. Myometrial samples from MED12 mutation-negative adjacent leiomyomas (Myo-WT) are shown in black and myometrial samples from MED12-mutated adjacent leiomyomas (Myo-Mut) are shown in red. (**D**) The protein–protein interaction networks were constructed with the Search Tool for the Retrieval of Interacting Genes (STRING) database and Cytoscape software version 3.9.1 using the 150 hub genes identified by the CytoHubba plugin of the Cytoscape software platform. The color of the nodes denotes interaction degree (from high to low degree: red, orange, yellow, and blue). (**E**) Gene ontology (GO) analysis of the identified differentially expressed 150 hub genes in the myometrium comparing the MED12-mutated (n = 11) with the non-mutated (n = 8) group of their paired adjacent leiomyomas (fold change ≥ 1.5, *p* < 0.05). Color gradient represents level of log2 fold change presented as z-scores.

**Figure 4 ijms-24-03742-f004:**
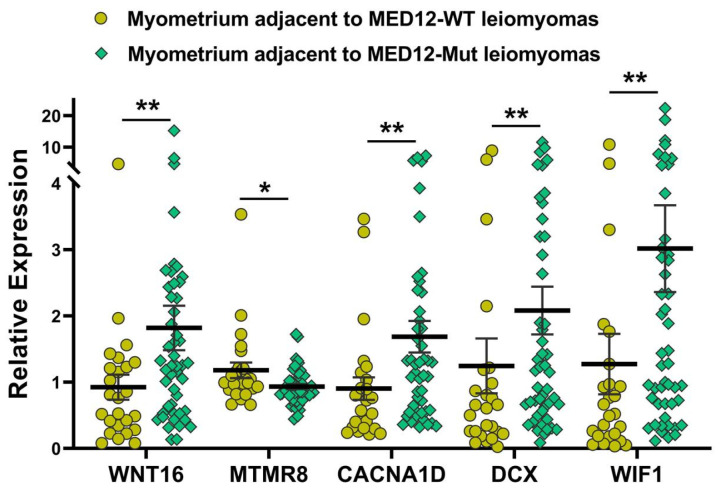
The expression of selected genes from the analysis of the myometrium (n = 73) based on MED12-mutation status of their paired adjacent leiomyomas by qRT-PCR. The results are presented as mean ± SEM with *p* values (* *p* < 0.05; ** *p* < 0.01) as indicated by the corresponding lines.

**Figure 5 ijms-24-03742-f005:**
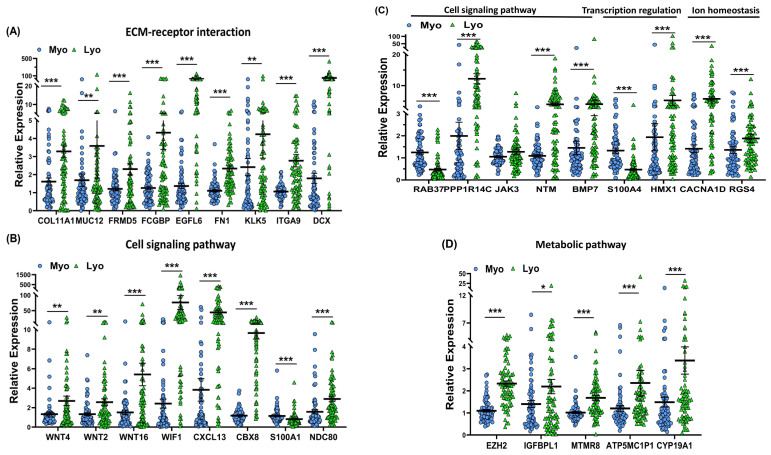
The expression of selected genes from the analysis between myometrium and paired leiomyomas (n = 73) measured using qRT-PCR. (**A**) *COL11A1*, *MUC12*, *FRMD5*, *FCGBP*, *EGFL6*, *FN1*, *KLK5*, *ITGA9*, and *DCX*; (**B**) *WNT4*, *WNT2*, *WNT16*, *WIF1*, *CXCL13*, *CBX8*, *S100A1*, and *NDC80*; (**C**) *RAB37*, *PPP1R14C*, *JAK3*, *NTM*, *BMP7*, *S100A4*, *HMX1*, *CACNA1D*, and *RGS4*; (**D**) *EZH2*, *IGFBPL1*, *MTMR8*, *ATP5MC1P1*, and *CYP19A1*. The results are presented as mean ± SEM with *p* values (* *p* < 0.05; ** *p* < 0.01; *** *p* < 0.001) indicated by corresponding lines.

**Figure 6 ijms-24-03742-f006:**
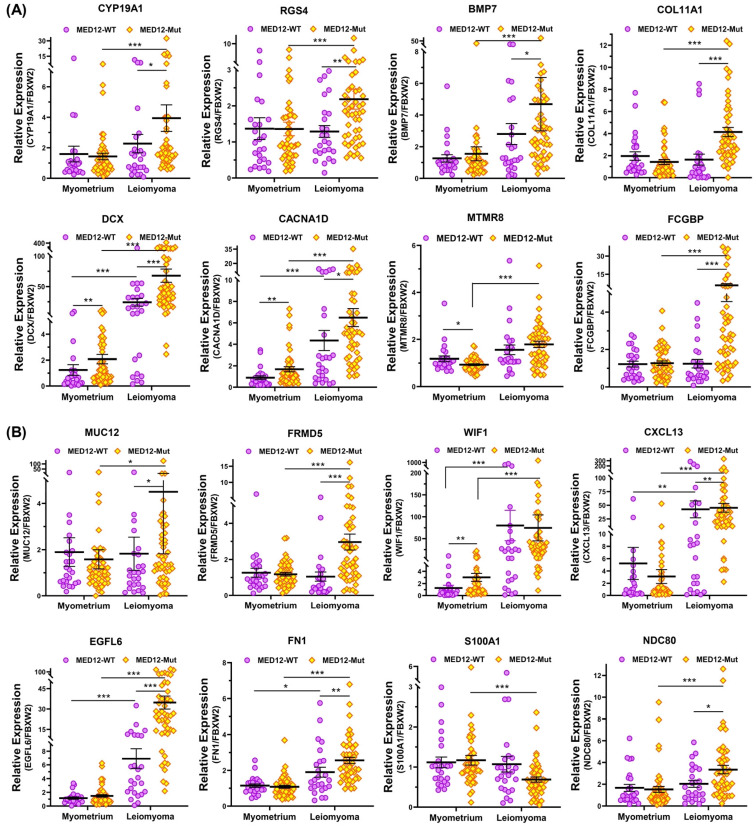
The expression of mRNA for (**A**) CYP19A1, RGS4, BMP7, COL11A1, DCX, CACNA1D, MTMR8, and FCGBP; (**B**) MUC12, FRMD5, WIF1, CXCL13, EGFL6, FN1, S100A1, and NDC80; (**C**) CBX8, KLK5, ITGA9, EZH2, RAB37, WNT4, WNT2, and WNT16; (**D**) PPP1R14C, JAK3, NTM, ATP5MC1P1, S100A4, HMX1, and IGFBPL1 in MED12-mutated leiomyomas (n = 48) and non-mutated leiomyomas (n = 25) and myometrium adjacent to MED12-mutated (n = 48) or non-mutated leiomyomas (n = 25) measured using qRT-PCR. The results are presented as mean ± SEM with *p* values (* *p* < 0.05; ** *p* < 0.01; *** *p* < 0.001) as indicated by the corresponding lines.

**Figure 7 ijms-24-03742-f007:**
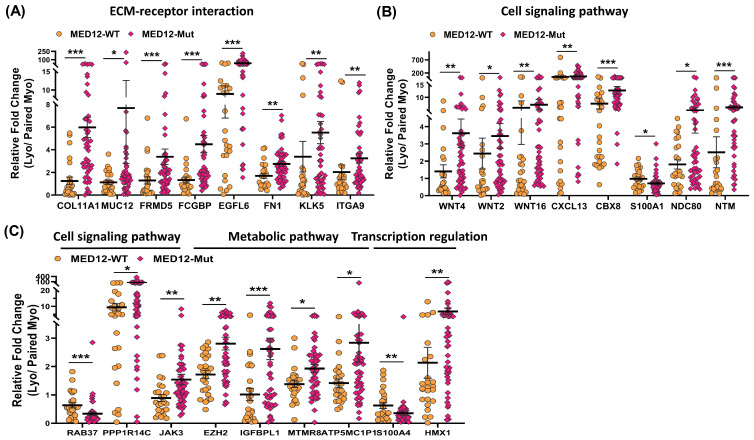
The expression of mRNA for (**A**) COL11A1, MUC12, FRMD5, FCGBP, EGFL6, FN1, KLK5, and ITGA9; (**B**) WNT4, WNT2, WNT16, CXCL13, CBX8, S100A1, NDC80, and NTM; (**C**) RAB37, PPP1R14C, JAK3, EZH2, IGFBPL1, MTMR8, ATP5MC1P1, S100A4, and HMX1 expressed as fold change (Lyo/paired Myo) in MED12-mutated (n = 48) and non-mutated (n = 25) specimens measured using qRT-PCR. The results are presented as mean ± SEM with *p* values (* *p* < 0.05; ** *p* < 0.01; *** *p* < 0.001) as indicated by the corresponding lines.

**Table 1 ijms-24-03742-t001:** Genes selected based on the RNAseq analysis according to the MED12 mutation status.

GO/KEGG Pathway Enrichment	Symbol	Lyo vs Myo	Lyo/Myo^(+)^ vs Lyo/Myo^(−)^	Lyo^(+)^ vs Lyo^(−)^	Myo^(+)^ vs Myo^(−)^	Function
ECM-receptor interaction	COL11A1	Up (*p* < 0.001)	Up (*p* < 0.001)	Up (*p* < 0.001)	No Significance	Minor fibrillar collagen; expression has been associated with advanced tumorigenic disease and epithelial-mesenchymal transition (EMT) [37].
ECM-receptor interaction	MUC12	Up (*p* < 0.05)	Up (*p* < 0.05)	Up (*p* < 0.05)	No Significance	An O-glycosylated protein of the mucin family; mucous barrier on epithelial surfaces; involves in adhesion modulation, epithelial renewal, differentiation and intracellular signaling via EGF-like domains in its extracellular region [38].
ECM-receptor interaction	FRMD5	Up (*p* < 0.001)	Up (*p* < 0.001)	Up (*p* < 0.001)	No Significance	Part of adherens junction and involved in regulation of cell migration, cellular metabolism, and signal transduction [39,40].
ECM-receptor interaction	FCGBP	Up (*p* < 0.001)	Up (*p* < 0.001)	Up (*p* < 0.001)	No Significance	Contains multiple von Willebrand D (VWD) domains that form complexes through disulfide-linked heterodimers with members of the mucin and trefoil factor family, which affect the attachment and motility of pathogens on mucosal surfaces [41,42].
ECM-receptor interaction	EGFL6	Up (*p* < 0.001)	Up (*p* < 0.001)	Up (*p* < 0.001)	No Significance	Member of EGF superfamily, is expressed at significant levels during developmental processes and various malignant cancers; involved in the regulation of cell cycle, tumor proliferation, invasion, and metastasis through activation of multiple signaling pathways including PI3K/AKT, ERK/MAPK, Wnt/β-catenin and integrin-mediated signaling pathway [43].
ECM-receptor interaction	FN1	Up (*p* < 0.001)	Up (*p* < 0.01)	Up (*p* < 0.01)	No Significance	A major component of the extracellular matrix, exists as a dimeric or multimeric form with other extracellular matrix proteins such as integrins, collagen, fibrin, and heparan sulfate proteoglycans linked by disulfide bonds [44,45].
ECM-receptor interaction	KLK5	Up (*p* < 0.01)	Up (*p* < 0.01)	Up (*p* < 0.001)	No Significance	KLK5, a member of the kallikrein subfamily, is involved in collagen formation and MSP-RON signaling [46,47].
ECM-receptor interaction	ITGA9	Up (*p* < 0.001)	Up (*p* < 0.01)	Up (*p* < 0.01)	No Significance	An integrin subunit that mediates cell-cell and cell-matrix adhesion and accelerates cell migration and regulates various biological functions including cancer cell proliferation, angiogenesis, adhesion and invasion [48].
ECM-receptor interaction	DCX	Up (*p* < 0.001)	No Significance	Up (*p* < 0.001)	Up (*p* < 0.01)	A microtubule-associated protein, contains two internal tandem repeats and stabilizes microtubules through bundling to the microtubule cytoskeleton [49].
Cell signaling pathway	WNT4	Up (*p* < 0.01)	Up (*p* < 0.01)	Up (*p* < 0.001)	No Significance	Ligand for members of the frizzled family of seven transmembrane receptors; works as a biphasic initiator for activating the canonical and non-canonical Wnt signaling [50].
Cell signaling pathway	WNT2	Up (*p* < 0.01)	Up (*p* < 0.05)	Up (*p* < 0.01)	No Significance	Enriched in cancer-associated fibroblasts; has the potential to enhance the growth and invasion of colorectal cancer [50].
Cell signaling pathway	WNT16	Up (*p* < 0.001)	Up (*p* < 0.01)	Up (*p* < 0.001)	Up (*p* < 0.01)	Has no homology to any other Wnts signaling molecule; implicated in tumorigenesis and in skeletal development and postnatal bone homeostasis [50].
Cell signaling pathway	CXCL13	Up (*p* < 0.001)	Up (*p* < 0.01)	Up (*p* < 0.01)	No Significance	Potent B lymphocyte chemoattractant, which promotes the migration of B lymphocytes, and is one of the most abundant chemokines in endometrial epithelial cells [51].
Cell signaling pathway	CBX8	Up (*p* < 0.001)	Up (*p* < 0.001)	Up (*p* < 0.001)	No Significance	Component of a PcG PRC1-like complex, is involved in the RNA polymerase II-mediated transcription repression of genes [52].
Cell signaling pathway	S100A1	Down (*p* < 0.001)	Down (*p* < 0.05)	No Significance	No Significance	Member of the S100 family of calcium-binding proteins; interacts with specific target proteins, resulting in the modulation of their activity [53,54].
Cell signaling pathway	NDC80	Up (*p* < 0.001)	Up (*p* < 0.05)	Up (*p* < 0.05)	No Significance	Component of the essential kinetochore-related NDC80 complex, which is involved in the organization and stabilization of microtubule-kinetochore interactions, spindle checkpoint signaling and chromosome segregation [55].
Cell signaling pathway	NTM	Up (*p* < 0.001)	Up (*p* < 0.001)	Up (*p* < 0.001)	No Significance	Neural cell adhesion molecule, promotes adhesion and neurite outgrowth via a homophilic mechanism [56].
Cell signaling pathway	RAB37	Down (*p* < 0.001)	Down (*p* < 0.001)	Down (*p* < 0.001)	No Significance	Rab small GTPase protein, through switching its guanine nucleotide binding status between GDP-bound (inactive) and GTP-bound (active) functions as a critical regulator in exocytotic pathway [57].
Cell signaling pathway	PPP1R14C	Up (*p* < 0.001)	Up (*p* < 0.05)	Up (*p* < 0.01)	No Significance	Inhibitor of the PP1 serine/threonine phosphatase, which regulates the activation of PTH(1–34)-induced catabolic response and the non-canonical PTH1R signaling pathway [58,59].
Cell signaling pathway	JAK3	No Significance	Up (*p* < 0.01)	Up (*p* < 0.01)	No Significance	Member of the JAK family of non-receptor tyrosine kinases, plays a pivotal role in cytokine and growth factor-mediated intracellular signal transduction via the JAK/STAT pathway [60].
Cell signaling pathway	WIF1	Up (*p* < 0.001)	No Significance	No Significance	Up (*p* < 0.01)	Secreted protein that binds and inhibits the activity of Wnt proteins.Downregulated in numerous cancers via epigenetic transcriptional silencing mechanism [61,62].
Cell signaling pathway	BMP7	Up (*p* < 0.001)	No Significance	Up (*p* < 0.05)	No Significance	Secreted ligand of the TGF-β superfamily, activates TGF-β signaling via receptor-mediated activation of Smad transcription factors leading to target genes regulation [63].
Metabolic pathways	EZH2	Up (*p* < 0.001)	Up (*p* < 0.01)	Up (*p* < 0.01)	No Significance	Catalytic core protein in PRC2 and facilitated PRC2-mediated H3K27me3 reaction. The dysregulation of EZH2 is acting as an important driver of tumorigenesis and progression [64].
Metabolic pathways	IGFBPL1	Up (*p* < 0.05)	Up (*p* < 0.001)	Up (*p* < 0.01)	No Significance	Located in extracellular space, is an IGF-binding protein that could prolong the half-life of IGF proteins and either promotes or inhibits the effects of IGF proteins on cell growth [65].
Metabolic pathways	MTMR8	Up (*p* < 0.001)	Up (*p* < 0.05)	No Significance	Down (*p* < 0.05)	Member of the myotubularin-related family; has phosphatase activity towards lipids containing phosphoinositol headgroup and acts on phosphatidylinositol 3-phosphate and phosphatidylinositol 3,5-bisphosphate; functions in membrane trafficking, cytoskeletal regulation, and receptor signaling [66].
Metabolic pathways	ATP5MC1P1	Up (*p* < 0.001)	Up (*p* < 0.05)	Up (*p* < 0.01)	No Significance	Pseudogene of ATP5MC1, which is encoded by the mitochondrial DNA and is subunit c of mitochondrial ATP synthase (F1F0 ATP synthase or Complex V) [67].
Metabolic pathways	CYP19A1	Up (*p* < 0.001)	No Significance	Up (*p* < 0.05)	No Significance	Member of the cytochrome P450 superfamily, is a monooxygenase involved in many reactions such as synthesis of steroids, lipids, cholesterol, and drug metabolism [68].
Transcription regulation	S100A4	Down (*p* < 0.001)	Down (*p* < 0.01)	Down (*p* < 0.01)	No Significance	Member of the S100 calcium-binding protein family, through its interaction with other proteins plays essential roles in many cellular processes and the development of cancers including metastasis, differentiation, inflammation, metastasis, and cell cycle progression [69].
Transcription regulation	HMX1	Up (*p* < 0.001)	Up (*p* < 0.01)	Up (*p* < 0.01)	No Significance	Transcription factor that belongs to the homeobox proteins (H6 family), and it recognizes and binds to the 5′-CAAG-3′ core DNA sequence. HMX1 may act as a transcriptional repressor and involved in the development of craniofacial structures [70].
Ion homeostasis	CACNA1D	Up (*p* < 0.001)	No Significance	Up (*p* < 0.05)	Up (*p* < 0.01)	Member of the high-voltage-activated Ca2+ channels (HVA), is expressed in uterus and involved in a variety of calcium signaling related processes [71].
Ion homeostasis	RGS4	Up (*p* < 0.001)	No Significance	Up (*p* < 0.01)	No Significance	Regulates numerous G protein-coupled receptors (GPCRs) associated post-receptor signaling cascades [72].

## Data Availability

Raw data were generated at The Lundquist Institute. Derived data supporting the findings of this study are available from the corresponding author O.K. on request.

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
