# Peer review of "Differential Expression of MED12-Associated Coding RNA Transcripts in Uterine Leiomyomas"

_ijms, 2023, doi:10.3390/ijms24043742_

Round 1
Reviewer 1 Report
In this work, the authors profiled 19 paired fibroid/myometrium samples (11 med12 positive, 8 med12 WT) and analyzed their transcriptomic output to generate meaningful observations. The manuscript is well written. Conclusions are supported by data. The following specific comments are provided.
Specific comments:
(1) Originality: The work is original and address an important topic in the field.
(2) The authors should comment on the ethnic/racial profile of the 19 OMICS samples. Also discuss any impact of ethnicity on their findings.
(3) The authors should discuss the significance of the observed changes between adjacent myometrium (MyoF) from Med 12 mutated versus wild type fibroids. Possibly the altered pathways in MyoF leading to med 12 mutagenesis such as increased oxidative stress are quite distinct from those leading to chemotrepsis (PMID: 35869560)
(4) The authors should comment how their data could explain the observation in prior literature that Med12 UF tends to be smaller in size and multiple in number versus med12 WT fibroids that tended to be solitary and larger.
Author Response
- Originality: The work is original and address an important topic in the field.
Response: We appreciate the comment in support of our work.
- The authors should comment on the ethnic/racial profile of the 19 OMICS samples. Also discuss any impact of ethnicity on their findings.
Response: Thanks for the suggestions. Due to the large amount of information on the racial differences of those coding genes, we are preparing another manuscript specific on this topic.
- The authors should discuss the significance of the observed changes between adjacent myometrium (MyoF) from Med 12 mutated versus wild type fibroids. Possibly the altered pathways in MyoF leading to med 12 mutagenesis such as increased oxidative stress are quite distinct from those leading to chemotrepsis (PMID: 35869560)
Response: Thanks for the suggestions. We have discussed the significance of the observed changes between adjacent myometrium and leiomyomas in the discussion section (line 250-267) and updated the new information as follows:
“The importance of the myometrium in the pathophysiology of leiomyomas was high-lighted in recent reports showing the myometrial oxidative stress can drive MED12 mutations in leiomyoma [77] and another report demonstrating differences in the transcriptome of myometrium from patients with fibroids as compared with myometrium from non-diseased uterus [78]. In addition, uterine contractions induce uterine hypoxia in reproductive-age women [79], and this insult can cause activation of NF-κB-mediated inflammation as has been demonstrated for human stromal cells [80].”
- The authors should comment how their data could explain the observation in prior literature that Med12 UF tends to be smaller in size and multiple in number versus med12 WT fibroids that tended to be solitary and larger.
Response: We do agree with the observation from prior reports regarding the correlation between the MED12 mutation status and the fibroids size and quantities; however, to reduce heterogeneity we limited our profiling to a single tumor within 3-5cm range and therefore we are unable to speculate about correlations between a particular MED12 mutation and tumor size or the number of tumors in a uterus.
Reviewer 2 Report
Chuang et al. studied the expression profile of coding RNA transcripts in leiomyomas with and without the MED12 mutation and their paired myometrium, presenting different comparisons between these four types of tissue.
There is evidence that demonstrated that exist differences between the transcriptomic profiles of myometrium adjacent to leiomyoma based on patient race (doi.org/10.1172/jci.insight.160274). Authors mentioned that “The paired tissues were from Caucasians (n=9), African Americans (n=23), Hispanics (n=37) and Asians (n=4)…”. How is the distribution of races in each of the four groups for RNA-seq and validation samples (myometrium adjacent to MED12-wt leiomyoma, myometrium adjacent to MED12-mut leiomyoma, MED12-wt leiomyoma, and MED12-mut leiomyoma)? Please add more details about this in the M&M section and discuss how it can affect the results observed.
There is evidence that a uterus can present several fibroids and they can be either WT or harbor MED12 mutation: “Of note there are several patients with multiple leiomyomas included in our study. Apparently, the different tumors from one patient can not only differ with respect to their cytogenetic subgroup but also to the occurrence and type of MED12 mutations thus representing further evidence for their independent clonal origin.” (DOI:10.1002/ijc.27424). Do the authors take this into account? Please discuss it.
Authors have found that “regulating response to oxygen containing compound” pathway is enriched in the adjacent myometrium to MED-12 mutant leiomyoma in comparison to the myometrium adjacent to MED12 wild-type leiomyomas. However, I found more interesting the enrichment of pathways related to inflammation more in this comparison (Figure 3E). Please, discuss it.
The authors used STRING database and Cytoscape software for the functional analysis of the protein-protein interactions network between leiomyomas compared to the myometrium (Figure 1D). It is not clear to me what information we can extract for this analysis since it is not very well discussed in the manuscript. For example: Are some of these associations already known to be involved in leiomyoma pathogenesis? Please consider this comment for Figure 3D as well (comparison between myometrium specimens based on the MED12 mutation status of their paired leiomyomas).
Figures 6. I suggest presenting in the main manuscript the most important/interesting results and presenting the rest of them as supplemental figures.
Table 1 could be a supplemental table.
Please, when referring to myometrium mention clearly that is the tissue adjacent to the leiomyoma. The terminology that the authors are using is sometimes confusing. For example, my suggestion is to change “Although we did not detect a mutation in MED12 gene in any myometrial specimen, we found marked differences in the expression profile in the myometrium of MED12 mutated specimens as compared with non-mutated ones” (Lines 121-123) to “Although we did not detect a mutation in MED12 gene in any myometrial TISSUES, we found marked differences in the expression profile in the myometrium ADJACENT TO MED12 mutated LEIOMYOMAS as compared with THE MYOMETRIUM ADJACENT TO MED12 WILD TYPE TUMORS”.
It seems that there is double spacing in Line 50- coactivator of / Line 51- factors controlling / Line 52- with MED13. If so, please correct them and check the manuscript for more.
Line 72. Please define the abbreviation of NGS since is the first time that appears in the introduction.
Figure 1B. Please, modify the Volcano plot to identify the distribution of the dots based on the significance and fold change (fold change >1.5 and P< 0.05) as in the Volcano Plot in Figure 3B.
Lines 103. Please, delete "following normalization of 29354 RNA transcripts" since this is already stated in line 83.
Line 112-113. “The heat map (Fig. 2B) shows 109 out of the 394 transcripts that were uniquely altered in the MED12 mutated specimens”. Why did the authors choose to show these 109 transcripts? Are these the top 109 uniquely altered or were chosen arbitrarily?
Line 114. KEGG abbreviation is defined in this line. Please delete the definition in lines 131 and 156.
Figure 4. Please, change the yellow circle and green square text legends to clarify that what is being analyzed are myometrium samples adjacent to leiomyomas (either MED12-mut or MED12-wt).
Figure 5. Indicate the letter of each panel in the text and put them in order as appeared in the figure, for example, line 158: ECM-receptor interaction (Figure 5A).
Line 166-167. “…which are all relevant to leiomyoma pathogenesis” please add references that support this affirmation (even if it is discussed in the discussion section later).
Line 172. “(Fig. 5A-E)”. Please correct it, figure 5 goes from A to D.
Line 174. “(Fig. 6A-F)”. Please correct it, figure 6 goes from A to D.
Figure 6 legend: “and mutated myometrium (n=48) and non-mutated ones (n=25)”. Please clarify that these are myometrium adjacent to MED12 mutated or non-mutated leiomyomas.
Lines 206 and 213-215. Authors mentioned: “We also identified a number of lncRNAs that were differentially expressed in the mutated samples for which there are no known functions currently” but there is no data regarding lncRNAs in this manuscript. Please, clarify/correct this.
Lines 261-264. Authors added references after a sentence that explains a result obtained in the study. lease, clarify/correct this.
Data Availability Statement: Add the Expression Omnibus (GEO) database accession number in this section once you have one.
Author Response
- There is evidence that demonstrated that exist differences between the transcriptomic profiles of myometrium adjacent to leiomyoma based on patient race (doi.org/10.1172/jci.insight.160274). Authors mentioned that “The paired tissues were from Caucasians (n=9), African Americans (n=23), Hispanics (n=37) and Asians (n=4)…”. How is the distribution of races in each of the four groups for RNA-seq and validation samples (myometrium adjacent to MED12-wt leiomyoma, myometrium adjacent to MED12-mut leiomyoma, MED12-wt leiomyoma, and MED12-mut leiomyoma)? Please add more details about this in the M&M section and discuss how it can affect the results observed.
Response: Thanks for the suggestions. Due to the large amount of information on the racial differences in these coding genes, we are preparing another report specifically on this topic.
- There is evidence that a uterus can present several fibroids and they can be either WT or harbor MED12 mutation: “Of note there are several patients with multiple leiomyomas included in our study. Apparently, the different tumors from one patient can not only differ with respect to their cytogenetic subgroup but also to the occurrence and type of MED12 mutations thus representing further evidence for their independent clonal origin.” (DOI:10.1002/ijc.27424). Do the authors take this into account? Please discuss it.
Response: We do agree with the statements from DOI:10.1002/ijc.27424; however, it is not feasible to test the type of mutation in a specimen with numerous fibroids. We only used one tumor from each patient, so we are unable to make any comment about the exact mutation type and the transcriptome profile of that specific tumor. We have added this sentence to reflect this limitation:
Line 378-380: “One limitation of our study is that a myomatous uterus has multiple tumors which might express different types of MED12 mutation [32], and these mutations may not induce the same changes in the transcriptome.”
- Authors have found that “regulating response to oxygen containing compound” pathway is enriched in the adjacent myometrium to MED-12 mutant leiomyoma in comparison to the myometrium adjacent to MED12 wild-type leiomyomas. However, I found more interesting the enrichment of pathways related to inflammation more in this comparison (Figure 3E). Please, discuss it.
Response: Thanks for the suggestions. We have discussed the significance of the observed changes between adjacent myometrium and leiomyomas in the discussion section (line 250-267) and updated new information as follow:
“The importance of the myometrium in the pathophysiology of leiomyomas was high-lighted in recent reports showing myometrial oxidative stress can drive MED12 mutations in leiomyoma [77] and another report demonstrating differences in the transcriptome of myometrium from patients with fibroids as compared with myometrium from non-diseased uterus [78]. In addition, uterine contractions induce uterine hypoxia in reproductive-age women [79], and this insult can cause activation of NF-κB-mediated inflammation as has been demonstrated for human stromal cells [80].”
- The authors used STRING database and Cytoscape software for the functional analysis of the protein-protein interactions network between leiomyomas compared to the myometrium (Figure 1D). It is not clear to me what information we can extract for this analysis since it is not very well discussed in the manuscript. For example: Are some of these associations already known to be involved in leiomyoma pathogenesis? Please consider this comment for Figure 3D as well (comparison between myometrium specimens based on the MED12 mutation status of their paired leiomyomas).
Response: Thanks for the suggestions. We have made some changes in the results section as follows:
Line 99-102: “ The involvement of many proteins such as FN1, COL1A1, BMP7 CCND1, EZH2, HMGA2, AhR and E2F1 shown in the protein-protein interaction networks are well known in fibroid pathogenesis [8, 28-32] and others such as SOX2, REN, PPARG, ABCB1 and NR3C1 are novel and require further investigation.”
Line 137-140: “Many proteins such as F3, WNT2, CDH1, LIF [33-36] shown in the protein-protein interaction networks are well known in fibroid pathogenesis and others such as ICAM1, CXCL8, CCL2 and NANOG are novel and require further investigation.”
- Figures 6. I suggest presenting in the main manuscript the most important/interesting results and presenting the rest of them as supplemental figures.
Table 1 could be a supplemental table.
Response: Thanks for the suggestions. With careful consideration we think the current way of presentation is easy and convenient for the readers.
- Please, when referring to myometrium mention clearly that is the tissue adjacent to the leiomyoma. The terminology that the authors are using is sometimes confusing. For example, my suggestion is to change “Although we did not detect a mutation in MED12 gene in any myometrial specimen, we found marked differences in the expression profile in the myometrium of MED12 mutated specimens as compared with non-mutated ones” (Lines 121-123) to “Although we did not detect a mutation in MED12 gene in any myometrial TISSUES, we found marked differences in the expression profile in the myometrium ADJACENT TO MED12 mutated LEIOMYOMAS as compared with THE MYOMETRIUM ADJACENT TO MED12 WILD TYPE TUMORS”.
Response: Thanks for the suggestions. We have made changes based on the request.
It seems that there is double spacing in Line 50- coactivator of / Line 51- factors controlling / Line 52- with MED13. If so, please correct them and check the manuscript for more.
Response: Thanks for the suggestions. We have made changes based on the request.
Line 72. Please define the abbreviation of NGS since is the first time that appears in the introduction. Response: Thanks for the suggestions. We have made changes based on the request.
Figure 1B. Please, modify the Volcano plot to identify the distribution of the dots based on the significance and fold change (fold change >1.5 and P< 0.05) as in the Volcano Plot in Figure 3B.
Response: Thanks for the suggestions. We build the Volcano plots of figure 1B and figure 3B all based on the significance and fold change (fold change >1.5 and P< 0.05).
Lines 103. Please, delete "following normalization of 29354 RNA transcripts" since this is already stated in line 83.
Response: Thanks for the suggestions. We have made changes based on the request.
Line 112-113. “The heat map (Fig. 2B) shows 109 out of the 394 transcripts that were uniquely altered in the MED12 mutated specimens”. Why did the authors choose to show these 109 transcripts? Are these the top 109 uniquely altered or were chosen arbitrarily?
Response: To visualize the results of the analysis we choose to present it in the format of heatmap. The 109 transcripts were chosen arbitrarily in order to show their symbol names clearly in the figure.
Line 114. KEGG abbreviation is defined in this line. Please delete the definition in lines 131 and 156. Response: Thanks for the suggestions. We have made changes based on the request.
Figure 4. Please, change the yellow circle and green square text legends to clarify that what is being analyzed are myometrium samples adjacent to leiomyomas (either MED12-mut or MED12-wt).
Response: Thanks for the suggestions. We have made changes based on the request.
Figure 5. Indicate the letter of each panel in the text and put them in order as appeared in the figure, for example, line 158: ECM-receptor interaction (Figure 5A).
Response: Thanks for the suggestions. We have made changes based on the request.
Line 166-167. “…which are all relevant to leiomyoma pathogenesis” please add references that support this affirmation (even if it is discussed in the discussion section later).
Response: Thanks for the suggestions. We have made changes based on the request.
Line 172. “(Fig. 5A-E)”. Please correct it, figure 5 goes from A to D.
Response: Thanks for the suggestions. We have made changes based on the request.
Line 174. “(Fig. 6A-F)”. Please correct it, figure 6 goes from A to D.
Response: Thanks for the suggestions. We have made changes based on the request.
Figure 6 legend: “and mutated myometrium (n=48) and non-mutated ones (n=25)”. Please clarify that these are myometrium adjacent to MED12 mutated or non-mutated leiomyomas.
Response: Thanks for the suggestions. We have clarified as “…. in MED12 mutated leiomyomas (n=48) and non-mutated leiomyomas (n=25) and myometrium adjacent to MED12 mutated (n=48) or non-mutated leiomyomas (n=25) by qRT-PCR.”
Lines 206 and 213-215. Authors mentioned: “We also identified a number of lncRNAs that were differentially expressed in the mutated samples for which there are no known functions currently” but there is no data regarding lncRNAs in this manuscript. Please, clarify/correct this.
Response: Thanks for the suggestions. We have fixed the mistakes.
Lines 261-264. Authors added references after a sentence that explains a result obtained in the study. lease, clarify/correct this.
Response: Thanks for the suggestions. We have added a line as “, which is in line with previous reports”.
Data Availability Statement: Add the Expression Omnibus (GEO) database accession number in this section once you have one.
Response: Thanks for the suggestions. Due to some technique issues, we still don’t have the accession number. We will update it once we receive.
Reviewer 3 Report
The objective of the study by Chuang and colleagues was to elucidate the expression profile of coding RNA transcripts in leiomyomas with and without MED12 mutations and their paired myometrium. Next generation RNA sequencing (NGS) was employed to profile both differentially expressed coding and non-coding RNA transcripts among the samples. To do so, 19 paired leiomyomas, including 8 MED12-mutation-negative and 11 MED12-mutation-positive leiomyomas were first evaluated, followed by 73 paired specimens for qRT-PCR validation of NGS data (48 with med12 mutation, 25 without). Overall, the study is clearly written, experiments are logical and outcomes will enhance our understanding on the pathophysiology of uterine fibroids. I have a few comments as outlined below.
11. For the RNA seq.(NGS) data what was the FDR cut-off value?
22. With respect to the data in Figure 4, there appears to be overlap in the relative expression of each of the 5 selected target genes, but are there any sample-specific characteristics (derived from AA patients, etc.) that show the greatest increased (for example, the samples that show greater than a 3-fold increase in MED12-mut for Wnt16, Wif1, etc?).
33. Although the sample size per race/ethnic group ranges from 4 to 37, would it be worthwhile to examine the association between the rate of MED12 mutation among these and correlation with the most differentially expressed genes of the 31 examined?
44. The authors indicate that they assessed 31 differentially expressed coding and non-coding RNAs by qRT-PCR in 73 paired samples. Could they indicate which of the listed genes were coding and non-coding? In the first line of the Discussion the authors mentioned lncRNAs, please indicate which if the 31 differentially expressed transcripts were lncRNAs.
55. Please increase the font size for the data in Table 1, perhaps re-format.
Author Response
- For the RNA seq.(NGS) data what was the FDR cut-off value?
Response: The FDR cut-off value as mentioned in the figure legends is 0.05
- With respect to the data in Figure 4, there appears to be overlap in the relative expression of each of the 5 selected target genes, but are there any sample-specific characteristics (derived from AA patients, etc.) that show the greatest increased (for example, the samples that show greater than a 3-fold increase in MED12-mut for Wnt16, Wif1, etc?).
Response: Thanks for the suggestions. Due to the large amount of information related to racial differences of the coding genes, we are preparing another report specifically on this topic.
- 3. Although the sample size per race/ethnic group ranges from 4 to 37, would it be worthwhile to examine the association between the rate of MED12 mutation among these and correlation with the most differentially expressed genes of the 31 examined?
Response: Thanks for the suggestions. Due to the large amount of information on the racial differences of those coding genes, we are preparing another manuscript specific on this topic.
- The authors indicate that they assessed 31 differentially expressed coding and non-coding RNAs by qRT-PCR in 73 paired samples. Could they indicate which of the listed genes were coding and non-coding? In the first line of the Discussion the authors mentioned lncRNAs, please indicate which if the 31 differentially expressed transcripts were lncRNAs.
Response: Thanks for the suggestions. We have fixed the mistakes.
- Please increase the font size for the data in Table 1, perhaps re-format.
Response: Thanks for the suggestions. We have made changes based on the request.